# Cytoprotective Effect of Liposomal Puerarin on High Glucose-Induced Injury in Rat Mesangial Cells

**DOI:** 10.3390/antiox10081177

**Published:** 2021-07-24

**Authors:** Lassina Barro, Jui-Ting Hsiao, Chu-Yin Chen, Yu-Lung Chang, Ming-Fa Hsieh

**Affiliations:** 1Department of Biomedical Engineering, Chung Yuan Christian University, Taoyuan 320, Taiwan; lassinabarro_dieron@yahoo.fr (L.B.); s0988041367@gmail.com (J.-T.H.); minimoney0628@gmail.com (C.-Y.C.); yulung.chang@gmail.com (Y.-L.C.); 2Department of Urology, Taoyuan General Hospital, Ministry of Health and Welfare, Taoyuan 320, Taiwan

**Keywords:** diabetic nephropathy, puerarin, liposome, renal mesangial cells, cytoprotective effect

## Abstract

In diabetic patients, high glucose and high oxidative states activate gene expression of transforming growth factor beta (TGF-β) and further translocate Smad proteins into the nucleus of renal cells. This signal pathway is characterized as the onset of diabetic nephropathy. Puerarin is an active ingredient extracted from *Pueraria lobata* as an anti-hyperglycemic and anti-oxidative agent. However, the poor oral availability and aqueous solubility limit its pharmaceutical applications. The present paper reports the liposomal puerarin and its protective effect on high glucose-injured rat mesangial cells (RMCs). The purity of puerarin extracted from the root of plant *Pueraria lobata* was 83.4% as determined by the high-performance liquid chromatography (HPLC) method. The liposomal puerarin was fabricated by membrane hydration followed by ultrasound dispersion and membrane extrusion (pore size of 200 nm). The fabricated liposomes were examined for the loading efficiency and contents of puerarin, the particle characterizations, the radical scavenge and the protective effect in rat mesangial cells, respectively. When the liposomes were subjected to 20 times of membrane extrusion, the particle size of liposomal puerarin can be reduced to less than 200 nm. When liposomal puerarin in RMCs in high glucose concentration (33 mM) was administered, the over-expression of TGF-β and the nuclear translocation of Smad 2/3 proteins was both inhibited. Therefore, this study successfully prepared the liposomal puerarin and showed the cytoprotective effect in RMCs under high glucose condition.

## 1. Introduction

Diabetes affects 422 million people worldwide and causes around 1.6 million deaths each year. Diabetes can be defined as a heterogeneous aetio-pathology that includes defects in insulin secretion, insulin action, or both, and disturbances of carbohydrate, fat, and protein metabolism [1]. It is a metabolic disease characterized by elevated levels of blood glucose. The chronicity of diabetes leads to severe complications such as cardiopathy, retinopathy, and neuropathy [2,3]. Diabetic nephropathy (DN) is the most common cause of end-stage renal disease and is a severe complication of diabetes [4]. This complication leads to end-stage kidney disease requiring hemodialysis and kidney transplantation to replace the failed kidney function. In addition, DN is associated with an increased risk of death in general [3]. 

Hyperglycemia induces renal tissue damage by involving several mechanisms. However, oxidative stress induced-hyperglycemia is the prominent phenomenon engaging the transforming growth factor beta1 (TGF-β1). The activated TGF-β1 promotes by phosphorus acidification, activates downstream transfer factors. TGF-β1 regulates the expression of gene transcription in the nucleus by activating downstream Smad-dependent and Smad-independent pathways [5]. Additionally, mitogen-activated protein kinase (MAPK) can also activate the Smad protein pathway through negative regulation. MAPK phosphatase causes the phosphorylation of Smad-dependent protein [6,7]; the metabolic pathways of advanced glycation end products (AGEs) and angiotensin II are caused by hyperglycemia that can activate the Smad-dependent protein the extracellular signal-regulated kinase (ERK)/p38 MAPK pathway. The protein complex enters the nucleus, and the DNA sequence is combined. It regulates the performance of gene transcription. The Smad-dependent pathway of the intracellular message transmission factor plays a significant role in the mechanism of diabetic pathology [8,9]. 

Rat renal mesangial cells (RMCs) have been used in several studies in high glucose conditions. Tang et al. found a high level of TGF-β1 and its downstream delivery factor Smad 2/3. They demonstrated that the high glucose environment promotes TGF-β1, activates the Smad 2/3 pathway, and allows cells to stay in the G0/G1 period in large numbers, causing a significant accumulation of extracellular matrix, the mechanism of DN [10]. They also demonstrated the effects of anti-oxidants as inhibitors of this accumulation of extracellular matrix. The anti-oxidants targeting the source of reactive oxygen species (ROS) generation prevent oxidative damage and subsequent progression of nephropathy [11,12].

Puerarin (8-(β-D-glucopyranosyl)-4′,7-dihydroxyisoflavone) is an isoflavone derived from Kudzu roots. The chemical structure of puerarin (Figure 1) contains two benzene rings, pyran and glucoside. The hydroxy groups bonded to different benzene rings lead to different components of isoflavones, e.g., daidzin, daidzein, genistin, genistein, and puerarin, respectively [13]. This isoflavone, isolated from the root of the plant *Pueraria lobata,* has anti-oxidant activities [14]. It is well-known that puerarin protects cells against oxidative stress [15]. Puerarin can decrease the expression of TNF-α and ameliorate insulin resistance in gestational diabetes mellitus rats by reducing glucose and lipid metabolism disorders [16]. However, as with other isoflavonoids, puerarin insolubility often limits its bioavailability. It may be one reason why many clinical studies have failed to find a positive association between isoflavone intake and the prevention of chronic diseases such as diabetes [17,18]. 

Nowadays, the particulate-based drug delivery system (DDS) have been extensively developed to resolve the poor solubility of drug candidates [19,20,21]. Among the DDS, liposomes are fabricated by cholesterol and phospholipids which are well validated by healthcare authorities around the world. In this research, puerarin isolated from the root of the plant *Pueraria lobata* was encapsulated in a liposome, and its cytoprotective effects against high glucose-induced injury in RMCs were investigated.

## 2. Materials and Methods

### 2.1. Raw Materials

Formic acid (FA), acetonitrile (AC, HPLC grade), dimethyl sulfoxide (DMSO) were obtained from Sigma-Aldrich (St. Louis, MO, USA). Raw materials for the preparation of the liposome, L-α-phosphatidylcholine, cholesterol, and D-α-tocopheryl polyethylene glycol 1000 succinate were purchased from Sigma-Aldrich. Reagents for in vitro experiments, 3-(4,5-dimethylthiazol-2-yl)-2,5-diphenyltetrazolium bromide (MTT), Triton X-100, D-(+)-glucose anhydrous, D-mannitol and 2,2-Diphenyl-1-picrylhydrazyl (DPPH) were purchased from Sigma-Aldrich. 2′,7′-Dichlorodihydrofluorescein diacetate (H_2_DCFDA) was purchased from Invitrogen. Methanol was purchased form Echo Chemicals Co., Ltd. (Miaoli, Taiwan).

### 2.2. Extraction and Characterizations of Puerarin

#### 2.2.1. Extraction of Puerarin from the Root of Plant *Pueraria lobata*

There are many methods for separating active ingredients from Chinese medicinal materials. However, the organic solvent method is the most convenient due to puerarin solubility in alcohol [22,23,24]. Briefly, a total of 10 g of *Pueraria lobata* root were first ground into powder, and then refluxed with 100 mL of 50% ethanol for two hours at room temperature, 50 °C and 70 °C, respectively. Thus, the effect of temperature on the extraction was evaluated. The crude products dissolved from *Pueraria lobata* was collected by filtering out of the solid residues. The rotary evaporator (N-1000, Eyela, Tokyo, Japan) was used to remove ethanol. To separate puerarin from the crude product, the dried powder was dissolved in 50 mL of acidic deionized water (pH = 4), and an equal volume of *n*-butanol was added into separatory funnel. After throughout mixing, the *n*-butanol was collected (approximately 45 mL). The *n*-butanol was throughout mixed with an equal volume of basic deionized water (pH = 8). When the organic phase and water phase were clearly separated, the water phase was collected. The recovered water was concentrated under reduced pressure and freeze-dried to obtain yellow-brown puerarin.

#### 2.2.2. UV/Visible Spectroscopy and High-Performance Liquid Chromatography for Purity of Puerarin Extract 

The ultraviolet/visible spectroscopy (UV-1800, Shimadzu, Japan) (UV/Vis) was performed to evaluate the purity of puerarin extract. A standard of puerarin (CAS number 3681-99-0) with the purity of 99.1%, recorded in the certificate of analysis, was obtained from MP Biomedicals (USA). To analyze the purity of puerarin extract, we performed UV/Vis spectroscopy with a wavelength range from 190 to 700 nm. A total of 1 mg of puerarin extract and standard substance were dissolved in dd H_2_O in a dark environment and prepared a serial dilution of puerarin solutions (20 to 70 μg/mL). The absorption maximum was found to be 250 nm (Figure 2). To further resolve the components of puerarin extract, high-performance liquid chromatography (Alliance HPLC system, Waters, Milford, MA, USA), equipped with an Atlantis C18 column (150 × 4.6 mm id, 5 μm) was employed. HPLC condition was adapted from previous works [25,26]. Briefly, 100 μL of the sample or standard (1 mg/mL) were injected into HPLC system and eluted with 0.1% formic acid (FA) and acetonitrile (AN) at a flow rate of 1.0 mL/min. The solvent gradients were 85% FA and 15% AN for 0 to 10 min and 30% FA and 70% AN for the rest of chromatogram (10 to 20 min). The column temperature was set at 30 °C. The eluent was monitored by the UV/Vis detector at a wavelength of 250 nm. The purity was then calculated from the area under the peaks of the puerarin extract, normalized to the area of all peaks.

#### 2.2.3. The Radical Scavenge Assay of Puerarin 

2,2-Diphenyl-1-picrylhydrazyl (DPPH) is a stable free radical with a strong absorption at a wavelength at 517 nm. In the present assay, puerarin scavenges DPPH radical by donating proton and results in oxidized form. The assay is to estimate the antioxidant activity of puerarin. The assay protocol was adapted from a comparative study of antioxidant activity of various puerarin sources [27]. Firstly, a calibration curve (absorbance versus concentrations) of DPPH was established using a UV/Vis spectroscopy. Secondly, 200 μM of DPPH solution was prepared in methanol, then mixed with various concentrations of puerarin (1.5, 1.25, 1.0, 0.75, 0.5, 0.25 mg/mL) at volume ratio of 1:2. The control is the one without addition of puerarin. Lastly, the absorbance of DPPH was measured at 517 nm using a UV/Vis Spectroscopy every 10 min upon the mixing. The DPPH scavenging activity of puerarin was calculated using the equation (1). A plot of DPPH scavenging activity versus reaction time was used to find the IC_50_, the concentration of Puerarin to reduce a half of DPPH radical.
(1)$$\mathrm{DPPH}\;\mathrm{scavenging}\;\mathrm{activity}=1-\left(\frac{\mathrm{absorbance}\;\mathrm{of}\;\mathrm{sample}\;\mathrm{at}\;\mathrm{different}\;\mathrm{reaction}\;\mathrm{time}}{\mathrm{absorbance}\;\mathrm{value}\;\mathrm{of}\;\mathrm{control}}\right)\;\times \;100\%$$

#### 2.2.4. Preparation and Characterizations of Liposomal Puerarin

In this experiment, liposomes were prepared by membrane hydration [28], and then the liposomes were stored in sucrose solutions (iso-osmotic or isotonic solution). The raw materials of the liposome included L-α-phosphatidylcholine, cholesterol and D-α-tocopheryl polyethylene glycol 1000 succinate. To prepare the liposome, 5 mL of chloroform were used to dissolve different molar ratio of the raw materials (Table 1) and then dried in a round flask under reduced pressure at 30 °C. The membrane was then hydrated with 2 mL of puerarin aqueous solution (1 mg/mL) in ultrasonic bath at 50 °C to get as-prepared liposome solution. To constraint the particle size of the liposome, a mini-extruder (Avanti Polar Lipid Inc., Alabaster, AL, USA) with two polycarbonate membrane filters with pore size of 200 and 400 nm were used to squeeze the solutions for 20 times. Thus the liposome was forced to pass through a porous membrane and self-assembled into uniform nanoparticles. 

The methods for the determination of the loading efficiency and contents of puerarin in the liposomes were adapted from our previous paper [19]. The liposomal puerarin (approximately 2 mL) was placed in a sealed dialysis bag (the molecular weight cut-off MWCO = 1000). The bag was placed in 40 mL of isotonic sucrose solution for the removal of un-encapsulated puerarin. After equilibrium of dialysis, the sucrose solution was measured for un-encapsulated puerarin by UV/Vis spectroscopy. The loading efficiency of puerarin was then calculated by the Equation (2):(2)$$\mathrm{The}\;\mathrm{loading}\;\mathrm{efficiency}\;(\%)=\left(\frac{\left(\mathrm{original}\;\mathrm{Puerarin}\;-\;\mathrm{un-encapsulated}\;\mathrm{Puerarin}\right)}{\mathrm{original}\;\mathrm{Puerarin}}\right)\;\times \;100\%$$
where the original puerarin was 1 mg/mL. To estimate the loading contents, defined in the Equation (3):(3)$$\mathrm{The}\;\mathrm{loading}\;\mathrm{contents}\;(\%)=\left(\frac{\mathrm{encapsulated}\;\mathrm{Puerarin}}{\mathrm{Phosphorous}\;\mathrm{concentration}\;\mathrm{of}\;\mathrm{liposome}}\right)\;\times \;100\%$$
where the numerator of the Equation (3) is equivalent to the numerator of the Equation (2). For the denominator (phosphorous concentration of liposome), the colorimetric method was conducted [29]. Briefly, the phosphate ionic group in the liposome was converted into inorganic phosphorus, followed by the salt of molybdenum ammonia phosphate, and lastly, molybdenum blue in which the absorbance was recorded by UV/Vis spectroscopy. 

#### 2.2.5. Particle Size Analysis of Liposomal Puerarin

A dynamic light scattering analyzer (3000HAS, Zetasizer, Malvern, PA, USA) was employed to measure the particle size of liposomal puerarin. An isotonic sucrose solution was used to dilute the liposomes to a final concentration of 0.5 mg/mL. The experimental conditions were set up at 25 °C, light scattering angle at 90° and the laser wavelength at 633 nm. A zeta potential module equipped in the Zetasizer was used to determine the surface charge of the liposomes using the electrophoretic scanning for 5 times. 

#### 2.2.6. Morphology Observation of the Liposome under Transmission Electron Microscope (TEM)

The liposome sample was prepared by a negative staining method. Phosphotungstic acid (PTA) was deposited on the periphery of the liposome, making the liposome and its background a strong contrast. Then, 0.2 g of PTA powder was dissolved in 20 mL ddH_2_O in a dark environment. A total of 5 µL liposome solution was dropped on the copper mesh and dried for 5 min. Then, 5 µL of PTA were then added and incubated for 8 min to allow heavy metals to settle around the liposome. A TEM (JEM-1400, JOEL, Tokyo, Japan) was employed to observe the morphology of the liposome.

### 2.3. Cytoprotective Effect of Iposomal Puerarin on RMCs 

#### 2.3.1. Normal Cell Culture and the High Glucose-Induced Injury of RMCs

The rat renal mesangial cells (RMCs) were acquired from ATCC (American Type Culture Collection, Rockville, MD, USA) [30,31]. After being unfrozen from liquid nitrogen, the RMCs were cultured in DMEM (Dulbecco’s modified eagle’s medium)/high glucose supplemented with 10% fetal bovine serum (FBS) and 1% antibiotics (penicillin-streptomycin, PS) in an incubator kept at 5% CO_2_, 37 °C and 100% humidity. By counting the cell numbers with a hemocytometer, the doubling time of RMCs was found to be 18.7 h for monitoring a stable cell culture of RMCs.

High glucose-induced injury of RMCs was established as in vitro diabetic model of RMCs in the present study. RMCs were seeded at a density of 3 × 10^3^ cells/well in 48 wells plate with different concentration of glucose medium (25, 33, 56 mM), and mannitol medium (25, 33, 56 mM) as the negative control group (to rule out the effect of osmotic pressure on rapid proliferation of RMCs). 

#### 2.3.2. Cell Viability Assay for Puerarin and Its Liposome

To find the non-toxic concentrations of puerarin extract and the liposomes, MTT assay was employed for the cell viability. For MTT stock solution, 500 mg of 3-[4,5-dimethylthiazol-2-yl]-2,5-diphenyltetrazolium bromide (MTT reagent) powder was dissolved in 100 mL PBS. The solution was filtered by 0.22 mm filter and stored in the refrigerator at 4 °C. The RMCs were seeded in a 96-well cell dish at a density of 2 × 10^3^ cells/well. After 24 h incubation, the medium was removed and the planted cells were washed twice. The medium were refreshed with new medium containing puerarin extract, empty liposome (placebo), and liposomal puerarin at the concentrations of 10, 25, 50, 100, and 200 µM, respectively. After 24 h of cultivation, the media were removed and replaced with medium/MTT solution mixed at the ratio 4:1, and incubated for 4 h protected from light. At the end of this incubation, the MTT solution was removed and replaced with DMSO solution for 5 min, and then the absorbances were measured by ELISA reader at 570 nm wavelength. The cell survival rate was calculated from the following Formula (4).
(4)$$\mathrm{Cell}\;\mathrm{survival}\;\mathrm{rate}=\left(\frac{\mathrm{absorption}\;\mathrm{value}\;\mathrm{of}\;\mathrm{sample}\;}{\mathrm{absorption}\;\mathrm{value}\;\mathrm{of}\;\mathrm{control}\;\mathrm{group}}\;\right)\;\times \;100\%$$

#### 2.3.3. The Effect of High Glucose Environment on Intracellular ROS and of RMCs

We cultured the RMCs in high glucose medium at the density of 8 × 10^3^ cells/well in a 48-well plate to mimic the high blood glucose condition. H_2_DCFDA was used to investigate the effect of high glucose on the expression of intracellular ROS. The H_2_DCFDA can freely pass through the cell membrane, enter the cell, and react with esterase to produce H2DCF product that cannot penetrate the cell membrane. Then it is oxidized by the reactive oxygen species inside the cell to form the fluorescent dichlorofluorescein (DCF). By detecting fluorescent performance, the production of ROS in RMCs was measured. The ROS detected by this reagent includes ONOO^−^, H_2_O_2_, and ROO^•^. After dissolving H_2_DCFDA with DMSO at the concentration of 10 mM, we experimented with the manufacturer recommendation. The fluorescence dye was measured at the end of the reaction (Excitation: 485 nm; Emission: 538 nm). The data was normalized from the total protein (bicinchoninic acid assay) of RMCs for each culture condition.

#### 2.3.4. The Effect of High Glucose Environment on the Proliferation of RMCs

RMCs were seeded at a density of 3 × 10^3^ cells/well in 48 wells plate. The culture medium was supplemented with different concentrations (5.6 mM and 33 mM) of glucose or mannitol. In addition, the respective concentration of puerarin or liposomal puerarin (10, 25, 50, and 100 μM) was added to explore liposomal puerarin on the promotion of cell proliferation and the protective effect in a high-glucose environment. 

#### 2.3.5. The Cytoprotective Effect of Liposomal Puerarin on the TGF-β1 Gene Expression

RMCs were seeded on a 10 cm culture dish at a density of 6.5 × 10^5^ cells/dish. The medium was supplemented with 5.6 mM glucose medium, 33 mM glucose medium, and 33 mM mannitol medium. In addition, the different concentrations of puerarin or liposomal puerarin were added for the assessment. After 96 h of culture, RMCs were harvested by trypsinization. The PureLink^TM^ RNA mini kit (Invitrogen, Carlsbad, CA, USA) was used for RNA extraction and purification. The reverse transcriptase-polymerase chain reaction (RT-PCR) was used to detect the gene expression of TGF-β1 (the primers in Table 2). First, we used mRNA as a template to copy complementary DNA (cDNA) by reverse transcriptase. Then, we used a polymerase chain reaction to multiply the cDNA template to synthesize the target DNA fragment.

#### 2.3.6. Immunofluorescence Staining: Intracellular Localization of Smad Proteins 2/3

RMCs were seeded in a medium containing 33 mM glucose at a density of 4 × 10^3^ cells/well, in a 48-well plate. The cells were then treated with 25 μM of liposomal puerarin. After culturing for 3 days, the medium was removed, and the cells were washed twice with 1× PBS, followed by fixing with 3.7% formaldehyde at room temperature for 30 min. We incubated RMCs for 10 min with 0.1% Triton X-100, followed by 5% BSA solution for 30 min. After removing BSA solution, Rabbit polyclonal Smad 2 and Rabbit polyclonal Smad 3 antibodies (1:199 ratio) in PBS were added to the respective well and then incubated at 4 °C overnight. After the reaction at room temperature for 1 h, we removed the primary antibodies, and then washed twice with 1× PBS. The secondary antibodies (Alexa fluor^®^ 488 Goat anti-rabbit IgG) solution was added and protected from light and incubated at room temperature for 1 h. For nucleus staining, 10 µg/mL of Hoechst 33,258 in PBS were added in each well for 20 min of incubation. Smad 2 and Smad 3 proteins were observed with a fluorescent microscope (Nikon Eclipse Ti-S, Tokyo, Japan).

#### 2.3.7. Statistical Analysis

We used Microsoft Excel to calculate the experimental data, expressed as the mean ± standard deviation, and each test has at least 3 repetitions. After the data is integrated, a *T*-test/ANOVA is used for statistical analysis to determine whether the experimental data is statistically significant. For example, When the *p*-value is less than 0.05, it means there is a statistically significant difference (* represents the *p*-value is less than 0.05).

## 3. Results

### 3.1. Extraction and Characterization of Puerarin 

The extraction yield and the purity of puerarin are expressed in Table 3. The extraction process at room temperature had the highest purity 83.4 ± 2.8% compared to those of 70.5 ± 5.1 and 55.6 ± 4.3% at 50 °C and 70 °C, respectively. The puerarin standard product and puerarin extract were analyzed by UV/Vis. The results in Figure 2A shows that the maximum absorption of the puerarin standard was at 250 nm. Figure 2B shows the UV/Vis spectra of puerarin extract. According to the chemical structure of puerarin (Figure 1), the conjugated structure composed of the cinnamoyl group and the benzoyl group gave two distance UV absorption bands, e.g., a peak I at 300–380 nm and peak II at 220–280 nm.

Figure 2C displays the HPLC chromatograms of puerarin standards. The pure standard was precisely eluted at 3.6 min in the chromatogram. In Figure 2D, the chromatogram shows that the retention time of puerarin extract was 3.6 min. Therefore, the area under the peak was generated to analyze the purity of the puerarin extract (Table 3). Puerarin is a flavonoid that the hydroxyl group on the structure can provide protons to stabilize the free radical chain reaction. The DPPH assay in Figure 2D shows that the puerarin extract having the concentration of 1.25 and 1.5 mg/mL can reach a free radical scavenging activity of 20–30% within 10 min. The 50% inhibition concentration (IC_50_) was 1.5 mg/mL, and the scavenging time was within 50 min. Accordingly, the scavenging activity was concentration and time-dependent.

### 3.2. Preparation and Characterizations of Liposomal Puerarin

In the present study, the preparation procedures of the liposome were divided into two major steps. The first step utilized membrane hydration to obtain as-prepared liposome. However, the particle size and the size distribution were undesirable. Therefore, the second step of ultrasonic bath and membrane extrusion were carried out. Table 4 shows the effect of ultrasonic bath and membrane extrusion on the average particle size of empty liposome (placebo). The average particle sizes of placebo after ultrasonic bath were 916.3, 795.3 and 637.0 nm, respectively. When the porous membrane having the pore size of 200 nm was used, the liposomes subjected to 10 times of the extrusion resulted in the average particle size of 220.4, 197.3 and 183.0 nm, respectively. When the extrusion times increased to 20 times, the particle size slightly decreased. The particle size of liposome obtained by extrusion was reduced 3.48–4.16 times, compared with that of ultrasonic bath. 

Aside from the effectiveness of membrane extrusion on the particle size of the liposome, the morphology of the liposome (E_60_C_30_T_10_) was observed under TEM. Figure 3 shows the empty liposomes filtered through porous membrane having the pore size of 400 nm (Figure 3A) and 200 nm (Figure 3B). The shape of liposomes remained mostly spherical and the particles were not damaged after passing through the membrane. The particle size of liposomes in the Figure 3 was calculated by the freeware ImageJ. The results show that liposomes passed through the membrane of 400 nm had an average diameter of 672.2 ± 126.3 nm. Liposomes filtered through the membrane of 200 nm had an average diameter of 158.3 ± 13.44 nm.

The preparation of liposomes consists of a conventional membrane hydration followed by mechanical methods. To homogenize the liposomes, a step of ultrasonic bath at 50 °C was conducted for 60 min. Afterward, the extrusion of the liposomes through a porous membrane was conducted for the final liposomes. Based on the results described in the previous paragraph, the desired procedures of liposome preparation include membrane hydration, ultrasonic bath and membrane extrusion. 

To fabricate liposomal puerarin, the extract was added during the membrane hydration. Table 5 shows the loading efficiency and contents of puerarin in the liposomes. During the step of the membrane extrusion, all of the liposomes were subjected to 20 times of extrusion through a porous membrane of 200 nm. Except for E_70_C_30_T_0_, the average particle size can be decreased to less than 200 nm. The loading efficiency increased when the molar ratio of D-α-tocopheryl polyethylene glycol 1000 succinate (T) increased from 0 to 10 mol%. Similarly, the loading contents were also increased when the molar ratio of T increased. Apparently, the hydrophilicity of liposome components, e.g., the molar ratio of T plays a significant role in the quality of the liposomal puerarin. This result can be attributed to the poor water solubility of the puerarin extract. The addition of hydrophilic lipids in the preparation of liposome enabled higher loading efficiency of the puerarin extract and decreased the particle size of liposomes.

Figure 4 displays the stability of the liposomal puerarin for a period of seven days. The particles sizes of the liposomes remained roughly constant throughout the period. However, the surface charges changed apparently. When the molar ratio of tocopheryl derivative (T) increased from 0 to 10 mol%, the surface charge of the liposome decreased among three liposomes. This phenomenon further evolved during the incubation period, e.g., the charges were more negative when incubation time passed. It can be attributed to the hydroxy group of puerarin dissociated in the water inducing negative charges on the surface of liposome. The higher puerarin loaded in the liposome, the more negative the zeta potential [28]. 

### 3.3. Cytoprotective Effect of Puerarin in RMCs under High Glucose-Induced Injury 

The experiments started from the searching of non-toxic concentrations of puerarin extract and liposomal puerarin in the range of 0 to 200 µM. Figure 5A represents the viability of RMCs performed in MTT assay. At a concentration of 200 µM, liposomal puerarin presented better cell viability (90%) than that (65%) of puerarin extract. Beyond 100 µM of puerarin, there was a significant difference between them (*p* < 0.01). Therefore, the concentrations for cytoprotective experiments were determined to be less than 100 µM.

In the high glucose-induced injury model, the culture media were supplemented with three glucose (or D-mannitol) concentrations of 25, 33 and 56 mM to induce ROS within RMCs cultured for three days. Figure 5B shows the intracellular ROS, expressed as relative fluorescent unit (RFU) of 322.07 and 388.90 RFU/mg for 33 and 56 mM of glucose after three days of culture, which was 1.23–1.53 times higher than the average expression of ROS in the control (5.6 mM of glucose) of 261.33 RFU/mg, (*p* < 0.05). It displayed that concentration of glucose higher than 33 mM can activate inflammatory reaction in the RMCs, while no elevation of ROS of RMCs was observed when D-mannitol was added in the culture medium.

The effect of a high glucose environment on the cell proliferation ability of RMCs is shown in Figure 5C. Addition of 33 mM of glucose in the medium can lead to higher proliferation ability of RMCs, compared with the condition of normal glucose (control). Interestingly, no elevation of proliferation activity was observed when much higher glucose concentration (56 mM) was used. We defined that glucose concentration of 33 mM can result in the oxidative injury of RMCs displaying the elevation of ROS and high proliferation activity [10].

To observe the cytoprotective effect of puerarin extract and liposomal puerarin, RMCs were cultured in a 48-well plate at a density of 2000 cells/well, then puerarin extract and liposomal puerarin, having the concentrations of 10, 25, 50, 100 µM were added. Figure 5D–E show the proliferation ratios of puerarin extract and liposomal puerarin, respectively. Both figures indicated dose-dependent proliferation ratios of RMCs, with respect to the concentration of puerarin. In a comparison of the protection efficacy between puerarin extract (free drug) and liposomal puerarin, 50–100 µM of liposomal puerarin in Figure 5E were more effective to lower the proliferation ratio than that of the control (Figure 5E). 

### 3.4. Effects of Liposomal Puerarin on the Inhibition of TGF-β Expression and Translocation of Smad Proteins

The Figure 6A represents the electrophoresis of TGF-β1 gene and the internal control β-actin extracted from RMCs in different culture condition, while the Figure 6B is its quantitative expression. A total of 33 mM of high glucose medium significantly promoted the over-expression of the TGF-β1 gene in RMCs, which was significantly increased by 18.09% compared to the control (RMCs in 5.6 mM of glucose). The puerarin extract had minor effect on the suppression of TGF-β1 gene. However, when the culture media were supplemented with a concentration of 25 and 50 µM of liposomal puerarin, TGF-β1 gene expression was reduced (5th and 6th lanes in Figure 6A,B). 

The fluorescence staining of Smad 2 and 3 are shown in Figure 6C. In control where normal glucose (5.6 mM) was supplemented, the nucleus of RMCs (first row of Figure 6C) was stained with Hoechst 33258 (blue) and Smad 2/3 proteins in cytoplasm were stained with the fluorescent antibodies (green). When the glucose was as higher as 33 mM (high glucose-induced injury), Smad 2/3 proteins (green) were found to translocate into the nucleus (blue and green mixed in the nucleus, 2nd row of Figure 6C). That indicates a sign of the diabetic nephropathy in RMCs. When 25 mM of liposomal puerarin was co-administered to RMCs cultured in the injury status, the translocation of Smad 2/3 proteins was alleviated (3rd row of Figure 6C), indicating a cytoprotective effect in RMCs. 

## 4. Discussion

As we have mentioned in the Introduction section, puerarin has lower solubility in water and higher solubility in ethanol [22]. To isolate more isoflavones from the plant *Pueraria lobata*, solvent and temperature are two important parameters. In the first step of extraction (reflux), 50% ethanol solution was used. Concerning the temperature of reflux, we found that the extraction yield (Table 3) remained at 1.8–1.9%, irrespective of the temperature of reflux. As a result, the solvent plays a critical role in the extraction process. Looking into the HPLC analysis, the puerarin standard has the elution peak at 3.6 min (Figure 2C), while puerarin extract also exhibited most abundant component (puerarin) eluted at 3.6 min. Furthermore, the purity of puerarin extract calculated from HPLC chromatogram was 83.4% indicating that the process of the present study can give desirable purity and yield of puerarin. Of course, other methods to isolate puerarin can be considered. For example, column chromatography has been reported as a continuous process [14]. 

To overcome the low solubility of puerarin in water, the liposomal dosage form of puerarin was developed in the present study. To obtain as-prepared liposome, a dried membrane consisting of L-α-phosphatidylcholine, cholesterol, and D-α-Tocopheryl polyethylene glycol 1000 succinate was hydrated with puerarin aqueous solution. However as-prepared liposome structurally resembles the multilamellar vesicles, formed by the lipid self-assembly mechanism. To produce unilamellar vesicles (the desired structure of liposome), post treatments of ultrasound and membrane filtration were employed. The report provided more insights into the structure of liposome [32]. The lipid composition and the mechanical treatments can significantly affect the structure of liposome. In addition, the introduction of pegylated lipid can considerably change the lamellarity from 40 to 0 vol %. Similar result was also reported in our previous paper [19]. 

In this study, the membrane extrusion of as-prepared liposomes for more than 10 times was required, as shown in Table 4. In the formulations of the liposomes, the molar ratio of cholesterol was fixed, while that of L-α-phosphatidylcholine and D-α-Tocopheryl polyethylene glycol 1000 succinate were varied. For optimal results (Table 5), liposomal puerarin of E_60_C_30_T_10_ showed a desirable particle size of 199.4 nm and high loading efficiency of puerarin of 53.79%, which was chosen for the in vitro cell experiments. 

In the kidney, mesangial cells, along with mesangial matrix reside in the micro-vasculature of glomerulus. The functions of the filtration and excretion of kidney relies on the intact mesangial cells and the extracellular matrix. However, the high glucose of diabetic patients increases the oxidative stress in the glomerulus and subsequent signaling pathways. The elevation of oxidative stress is stemmed from the glycation of glucose on the cell membrane and followed by various metabolic events, such as polyol pathway, protein kinas C activation and advanced glycation end product synthesis [33]. Furthermore, the binding of TGF-β1 on the cell membrane has been discovered to signal its downstream pathway, e.g., Smad pathway in the cells. 

Tang et al. reported that around 62% of RMCs cultured in the medium containing 25 mM glucose for 48 h remain in G0/G1 phases, and the expression of TGF-β1 and its downstream genes, Smad 2/3 were 1.49 times more than RMCs cultured without glucose [10]. In a recent study, Mao et al. reported that triterpene saponins inhibited kidney fibrosis and the progression of DN. The administration of astragaloside IV, one of triterpene saponins, decreased the proliferation of high glucose-injured RMCs and the both mRNA and protein expressions of TGF-β1, Smad3, collagen I, α-smooth muscle actin in vitro and in vivo. Because the TGF-β1/ Smad3 pathway was blocked, the progression of renal fibrosis could be managed by those herbal drugs [34]. 

To mimic DN in an in vitro study, the cell line of rat mesangial cells (RMCs) was employed in the present study. It was found that RMCs cultured at 33 mM glucose led to high proliferation rate, as determined by MTT assay, while much higher glucose, e.g., 56 mM glucose didn’t further increase the proliferation as shown in Figure 5C. The high glucose culture conditions accelerate cells proliferation [35]. However, the behavior of RMCs in the 33 mM condition and 56 mM expressed two antagonist actions of TGF-β1 [36] by promoting the proliferation in the 33 mM glucose condition and inhibiting the proliferation in 56 mM glucose condition. It is expected that increased oxidative stress activates NF-kB pathway and leads to apoptosis of RMCs [37]. Meanwhile, the intracellular ROS in RMCs increased with the increase of glucose concentration. To rule out the effect of osmosis pressure on the cellular proliferation, D-mannitol of 25, 33 and 56 mM were used to culture RMCs. No statistical difference was noted in this control experiment, indicating that osmotic pressure didn’t affect the proliferation of RMCs (Figure 5B,C). To summarize the in vitro DN model, 33 mM glucose can increase the proliferation of RMCs and intracellular ROS, making this condition similar to the onset of DN.

Puerarin is categorized as an anti-oxidant obtained from *Pueraria lobata*. The present study aimed to provide liposomal puerarin which is designed to alleviate the oxidative stress inside the cells, in particular RMCs. When liposomal puerarin were added into the cell culture of RMCs in 33 mM glucose, the cell proliferation ratio decreased from 1.6 to 0.95. A dose-dependence between the dose of liposomal puerarin and cell proliferation ratio was observed as shown in Figure 5C. For comparison, free puerarin and its placebo (empty liposome) were examined to reduce the over-expression of TGF-β1 of RMCs in the in vitro DN model. We found that 50 mM liposomal puerarin performed much better than free puerarin (Figure 5D,E). 

When high glucose (33 mM) was administered, TGF-β1 signal pathway can trigger downstream events, in particular Smad protein translocation in cellular nuclei. In the fluorescent microscopic experiments shown in Figure 6C, the translocation of Smad 2/3 proteins into cellular nuclei was inhibited by the administration of liposomal Puerarin in in vitro DN model. The possible mechanism of such efficacy might be related to suppression of TGF-β1 signal pathway. Taken altogether, liposomal puerarin prepared in the present study was characterized in in vitro DN model for the suppression of RMC proliferation. Its molecular mechanism might be the inhibition of TGF-β1 pathway in which the downstream event of Smad protein translocation was stopped. 

The liposome is considered a ready-to-use dosage form since the ingredients such as phospholipids, cholesterol and tocopheryl derivatives. In a recent work, Liu et al. designed liposomal puerarin as anti-inflammatory targeting vehicle for the therapy of cerebral ischemia-reperfusion injury [38]. They discovered that neutrophils can carry those liposomal puerarin to the wound site where massive inflammatory reactions are prevailing. That report shines potential applications of liposomal puerarin for the DN therapy. 

## 5. Conclusions

Puerarin extract and liposomal puerarin were characterized and then supplemented with medium for rat renal mesangial cells in high glucose concentrations. The liposomal puerarin, consisting of 60 mol% of L-α-phosphatidylcholine, 30 mol% of cholesterol and 10 mol% of D-α-Tocopheryl polyethylene glycol 1000 succinate, prepared by membrane extrusion had desired properties (particle stability and drug loading parameters). A total of 33 mM of high glucose medium induced rat renal mesangial cells as the in vitro diabetic nephropathy model, which was confirmed by the gene expression of TGF-β1 and the translocation of Smad 2/3 proteins in the nucleus of the cells. Liposomal puerarin having a concentration of 25–100 μM was demonstrated to inhibit that gene expression of TGF-β1 and the trans location of Smad 2/3 proteins. This study paved the way for further animal evaluations of liposomal puerarin.

## Figures and Tables

**Figure 1 antioxidants-10-01177-f001:**
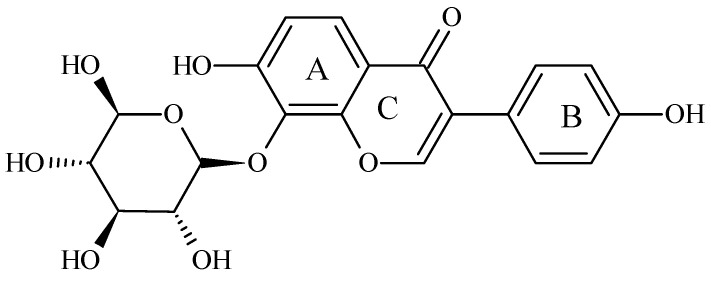
Chemical structure of puerarin. (IUPAC name: 8-(β-D-glucopyranosyl)-4′,7-dihydroxyisoflavone) [13].

**Figure 2 antioxidants-10-01177-f002:**
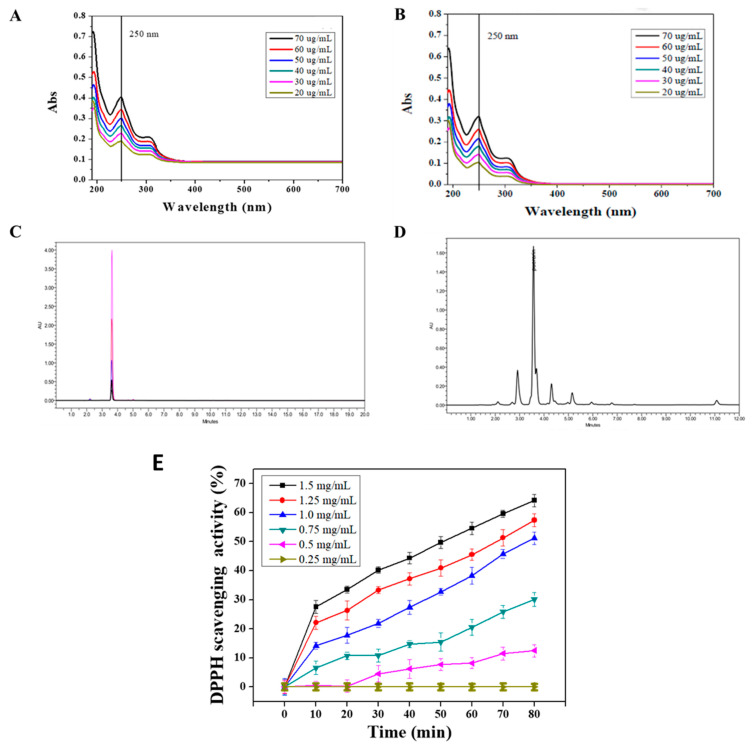
Characterizations of puerarin: (**A**,**B**) UV/Vis spectra of puerarin standard (CAS number 3681-99-0) and puerarin extract, (**C**,**D**) HPLC chromatograms of puerarin standard and puerarin extract, and (**E**) DPPH scavenging activity of puerarin extract.

**Figure 3 antioxidants-10-01177-f003:**
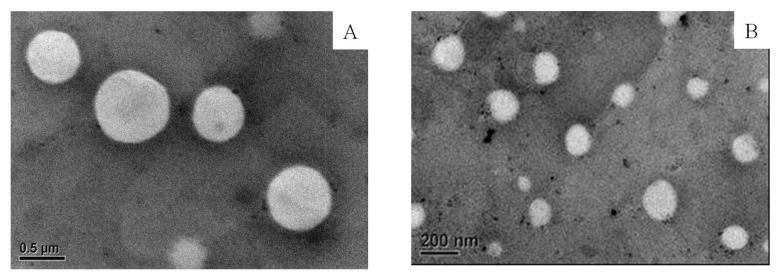
The TEM images of empty liposome (E_60_C_30_T_10_) filtered through the membrane of 400 (**A**) and 200 nm (**B**). The liposomes were stained with 1 wt % of PTA on carbon-coated cupper grid.

**Figure 4 antioxidants-10-01177-f004:**
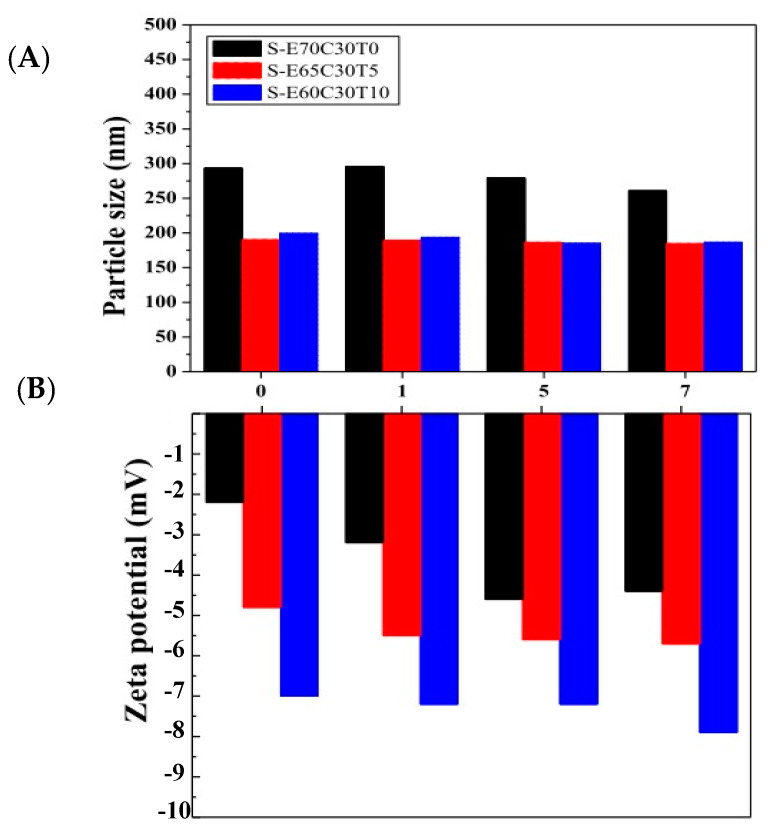
The stability of particle size and zeta potential of liposomal puerarin for a storage period of seven days. (**A**) the average particle size, (**B**) zeta potential.

**Figure 5 antioxidants-10-01177-f005:**
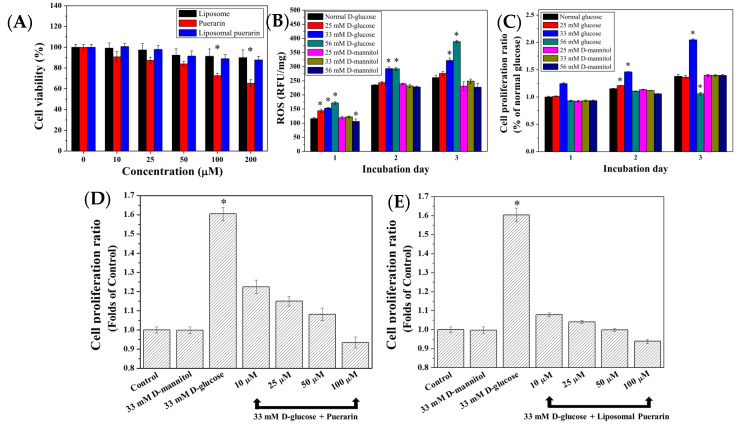
(**A**) Cell viability assay of empty liposome, puerarin extract and liposomal puerarin at various concentrations; (**B**) intracellular ROS of RMCs with respect to different glucose concentrations; (**C**) effects of glucose concentrations on proliferation of RMCs; (**D**,**E**) cytoprotective effect of puerarin extract and liposomal puerarin on RMCs in high glucose (33 mM). * *p* < 0.05. The ratios in Figure 5 (**C**–**E**) were calculated by normalizing with the control (RMCs in 5.6 mM of glucose).

**Figure 6 antioxidants-10-01177-f006:**
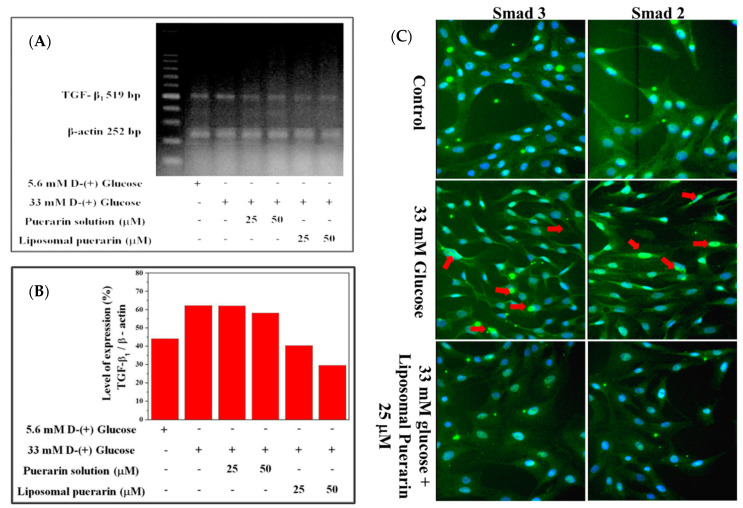
The effect of liposomal puerarin on TGF-β1 gene expression and the translocation of Smad 2/3 into the nucleus: (**A**) electrophorese of TGF-β1 gene extracted from RMCs; (**B**) quantitative expression of TGF-β1 gene; (**C**) fluorescence staining of Smad 2/3 of RMCs. Magnification: 400×. Control (1st row of Figure 6C): the nucleus of RMCs was stained with Hoechst 33258 (blue) and the cytoplasm was stained with antibodies of Smad 2/3 (green). When 33 mM of glucose were added, both Smad 2/3 proteins translocated in the nucleus, marked with red arrows (2nd row of Figure 6C). When 25 μM of liposomal Puerarin were used, the proteins were mainly found in cytoplasm (3rd row of Figure 6C).

**Table 1 antioxidants-10-01177-t001:** The compositions for the preparation of liposomes.

Liposomal Puerarin	Weight of E (mg)	Weight of C (mg)	Weight of T (mg)
E_70_C_30_T_0_	5.3	1.2	0
E_65_C_30_T_5_	4.9	1.2	0.8
E_60_C_30_T_10_	4.6	1.2	1.2

E, C and T stand for L-α-phosphatidylcholine, cholesterol and D-α-tocopheryl polyethylene glycol 1000 succinate, respectively. The numbers in the abbreviation of the liposome is the molar ratio of the raw materials.

**Table 2 antioxidants-10-01177-t002:** Primers of nucleic acid sequence and reaction parameters.

Genes	Access Number	Primers (Sense/Antisense)	Base Pair	Cycle
Rat TGF-β1	P17246	5′-CCCGCATCCCAGGACCTCTCT-3′	519	30
		5′-CGGGGGACTGGCGAGCCTTAG-3′		
β-actin	P60711	5′-GCTGCGTGTGGCCCCTGAG-3′	252	30
		5′-ACGCAGGATGGCATGAGGGA-3′		

**Table 3 antioxidants-10-01177-t003:** The effect of temperature on the yield and the purity of puerarin extract.

Temperature (°C)	Extraction Yield (%)	Puerarin Purity (%)
RT	1.82 ± 0.8	83.4 ± 2.8
50	1.90 ± 0.7	70.5 ± 5.1
70	1.80 ± 1.2	55.6 ± 4.3

RT: Room temperature (22–24 °C).

**Table 4 antioxidants-10-01177-t004:** Comparison of particle size of liposomes obtained by different methods.

Empty Liposome	Ultrasonic Bath	Membrane Extrusion (10 Times)	Membrane Extrusion (20 Times)
E_70_C_30_T_0_	916.3 nm	220.4 nm	202.8 nm
E_65_C_30_T_5_	795.3 nm	197.3 nm	186.4 nm
E_60_C_30_T_10_	637.0 nm	183.0 nm	165.1 nm

**Table 5 antioxidants-10-01177-t005:** The loading efficiency and contents of the puerarin in liposomes.

Liposomal Puerarin	Particle Size (nm)	Loading Efficiency (%)	Loading Contents (%)
E_70_C_30_T_0_	293.6	40.45	9.38
E_65_C_30_T_5_	190.3	49.24	9.56
E_60_C_30_T_10_	199.4	53.79	9.86

## Data Availability

The data presented in this study are available in article.

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
