# Peer review of "Cytoprotective Effect of Liposomal Puerarin on High Glucose-Induced Injury in Rat Mesangial Cells"

_antioxidants, 2021, doi:10.3390/antiox10081177_

Round 1
Reviewer 1 Report
This study tried to provide a better formulated antioxidant agent for the treatment of diabetic nephropathy. The authors demonstrated the whole fabrication processes of the active ingredient extraction from raw material, antioxidant potency analysis and the liposome formulation as well as the therapeutic functions of diminishing high glucose induced RMCs proliferation and TGF-B/SMAD signaling. Thought the idea of using liposome formulation to improve the potency of Puerarin has been proposed, the authors optimized the active ingredient extraction and liposomal formulation and showed good therapeutic potency in RMCs model. In addition, they showed that the anti-proliferation function of Puerarin and liposomal Puerarin is mediated by inhibiting the high glucose induced TGF-B/SMAD signaling. However, some issues make this study to be challenged.
- In most cell types, TGF-B signaling provoke growth inhibition machinery but it showed different functions in some oncogenic conditions (a). Similarly, the TGF signaling stimulates RMCs proliferation under high glucose condition. Thus, the authors are recommended to further discuss about this issue.
- The Y axis of Fig. 3C-3E seems to be mis-labeled.
- Since ref 10 was retracted and the content is similar to ref 11, the authors are recommended to remove ref 10.
a. TGF-β - an excellent servant but a bad master. Lenka Kubiczkova, Lenka Sedlarikova, Roman Hajek, Sabina Sevcikova. J Transl Med. 2012 Sep 3;10:183. doi: 10.1186/1479-5876-10-183
Author Response
see attached table

Reviewer 2 Report
In the original paper: "Cytoprotective Effect of Pegylated Liposomal Puerarin on High Glucose-Induced Injury in Rat Mesangial Cells," The present paper describes the preparation of pegylated liposomal Puerarin and its protective effect on high glucose-injured rat mesangial cells (RMCs). The problem presented in the manuscript is interesting because it shows attempts to modify puerarin to increase its biological thus therapeutic effect. Authors should check the article for linguistic correctness. Before publication, the manuscript needs to be improved, as there are many understatements when reading it. Below I present my suggestions, which will help to improve this manuscript.
Methods
- Preparation of the extract does not describe what part of the plant the extract was prepared from (point 2.1. - line 71) and what volume of solvent was poured over the raw material
- The term "Puerarin extract purity" or "puerarin extract" appears in work, which I do not fully understand. In section 2.1., the authors describe the preparation of Puerarin. There is no description in the paper regarding the preparation of "puerarin extract." Are they two identical substances? The authors should introduce the explanation.
- Has the extraction method been previously described and proven effective in terms of the compound that allows the equation? If not, better confirmation of the identity of the obtained substance would be required, e.g., MS.
- There is no "chemicals" paragraph to describe the reagents used in the work
- The authors in (paragraph 2.2.) Talk about the "standard" used - was it a commercially acquired puerarin? this information is missing
- In figure 1 A - does the spectrum represent puerarin extract or pure commercially acquired puerarin extract? If the extract (or isolated puerarin - it is not apparent to me from the description in the paper), the authors should also include the spectrum for the standard.
- Has the HPLC method used been used before somewhere? The authors do not cite any literature. If the method was developed for this work, it must be validated.
- For what purpose did the authors dissolve DPPH in various concentrations? I don't understand this course of action. Please explain?
- There is no information in the methodology on how the result will be expressed in the DPPH test. There is only half the information. It should also be mentioned which measurement point was selected as appropriate for calculating the IC50 parameter.
- The phrase "the control group" (line 109) is unfortunate, just "control."
- In the description of the HPLC methodology, the injection volume and the concentrations in which the substance was tested are missing
- Was the reference substance puerarin used in the HPLC analytical method? Authors must mark it.
- If the HPLC chromatogram (figure 1 C) is a chromatogram of the obtained Puerarin used for further research, the isolated compound is not pure. I conclude it is necessary to recalculate the percentage of this substance to determine with what amount we are dealing.
Discussion
- The first part of the discussion (lines 316-339) looks more like a summary of the methodology and results, giving repetition. Authors should change this by introducing elements of the discussion of these results based on literature reports.
Others
- Authors should introduce the puerarin structure as the next Figure
- Authors should check if the italic is used in the case of the latin name of “Pueraria lobata.”
- It should be: “Pueraria lobata,” not “Pueraria lobate” (line 1, line 316, 370)
- The Authors should introduce some more recent positions of the References
- puerarin is once write "Puerarin" (line: 81, 104 ...), secondly "puerarin" (line 75)
- Table 2 should be placed elsewhere - closer to the passage that describes it
- It should be: "in vitro" not "in-vitro" (line 377, 385, 355 etc.)
Author Response
see attached table

Reviewer 3 Report
The manuscript “Cytoprotective Effect of Pegylated Liposomal Puerarin on High Glucose-Induced Injury in Rat Mesangial Cell” is interesting and relevant for the field of antioxidant biology. However, there are some minor issues that need to be addressed in order to enrich the manuscript before publication.
1) the title with the term ‘pegylated liposomal puerarin” is unclear. Authors should careful and explain in the text the meaning of pegylated liposomes because it just relates to a one component and the generalization is not clear.
2) section 2.5: explain in detail how the encapsulated puerarin was measured. Also, explain how the loaded liposomes were separated from the extravesicular puerarin.
3) line 131: explain what is 0.5 mg/ml (saccharose? total lipids? Puerarin?).
4) section 2.8: describe how and where the rat renal mesangial cells were obtained.
5) section 2.9: specify MTT and its concentration in the assay.
6) editing issues: many places have issues with using sentence versus capital letters for same words (e.g. puerarin versus Puerarin, and many more); combine sections 2.11 and 2.12; revise Table 1 as it is weird to specify annealing temperature for single oligonucleotide primers; There are sentences with unclear structure, which need editing; use plural form with data; abbreviations should be explained once they are first mentioned (e.g. ROS), etc. The manuscript requires essential editing.
Author Response
see attached table

Round 2
Reviewer 2 Report
Thank you to the Authors for clarifying questionable points. In my opinion, the amendments introduced are sufficient. I find that in this form, the article can be published.